# The Influence of the Severity of Early Chronic Kidney Disease on Oxidative Stress in Patients with and without Type 2 Diabetes Mellitus

**DOI:** 10.3390/ijms231911196

**Published:** 2022-09-23

**Authors:** Jorge Andrade-Sierra, Leonardo Pazarín-Villaseñor, Francisco Gerardo Yanowsky-Escatell, Elodia Nataly Díaz-de la Cruz, Andrés García-Sánchez, Ernesto Germán Cardona-Muñoz, Francisco Javier Munguía-Galaviz, Alejandra de Alba-Razo, Alejandra Guillermina Miranda-Díaz

**Affiliations:** 1Nephrology Service, Department of Internal Medicine, Civil Hospital of Guadalajara “Dr Juan I. Menchaca”, Guadalajara 44340, Jalisco, Mexico; 2Specialty in Nephrology, Regional General Hospital No. 46 IMSS, Guadalajara 44220, Jalisco, Mexico; 3Department of Health Sciences-Illness as an Individual Process, University Center of Tonalá, Tonala 44430, Jalisco, Mexico; 4Department of Physiology, University Health Sciences Center, University of Guadalajara, Guadalajara 44340, Jalisco, Mexico

**Keywords:** chronic kidney disease, oxidative stress, diabetes mellitus, early chronic kidney disease, antioxidants, inflammation

## Abstract

Early Chronic Kidney Disease (CKD) is a condition that tends to progress to End-Stage Kidney Disease (ESKD). Early diagnosis of kidney disease in the early stages can reduce complications. Alterations in renal function represent a complication of diabetes mellitus (DM). The mechanisms underlying the progression of CKD in diabetes could be associated with oxidative and inflammatory processes. This study aimed to evaluate the state of inflammation and oxidative stress (OS) on the progression of CKD in the early stages in patients with and without type 2 diabetes mellitus (T2DM). An analytical cross-sectional study was carried out in patients with CKD in early stages (1, 2, 3) with and without T2DM. The ELISA method determined the expression of pro-inflammatory cytokines IL-6 and TNF-α as well as lipoperoxides (LPO), nitric oxide (NO), and superoxide dismutase activity (SOD). Colorimetric methods determined glutathione peroxidase (GPx) and total antioxidant capacity (TAC). Patients with CKD and T2DM had significantly decreased antioxidant defenses for SOD (*p* < 0.01), GPx (*p* < 0.01), and TAC (*p* < 0.01) compared to patients without T2DM. Consequently, patients with T2DM had higher concentrations of oxidant markers, NO (*p* < 0.01), inflammation markers, IL-6 (*p* < 0.01), and TNF-α than patients without T2DM. CKD stages were not related to oxidative, antioxidant, and inflammatory marker outcomes in T2DM patients. Patients without T2DM presented an increase in SOD (*p* = 0.04) and a decrease in NO (*p* < 0.01) when the stage of CKD increased. In conclusion, patients with T2DM present higher levels of oxidative and inflammatory markers accompanied by a decrease in antioxidant defense. However, these oxidative status markers were associated with CKD stage progression in patients without T2DM. Thus, NO and SOD markers could help detect the early stages of CKD in patients who have not yet developed metabolic comorbidities such as T2DM.

## 1. Introduction

Chronic kidney disease (CKD) is defined by the estimated glomerular filtration rate (eGFR) < 60 mL/min/1.73 m^2^ or by one or more markers of persistent kidney damage for three months. Albuminuria (albumin excretion rate ≥ 30 mg/day, albumin–creatinine ratio (ACR) ≥ 30 mg/g, urinary sediment abnormalities, electrolyte abnormalities, tubular disorders, histologically detected abnormalities, and imaging-detected structural abnormalities also define CKD. CKD is associated with decreased renal function related to increasing age and the presence of hypertension, diabetes mellitus (DM), obesity, and primary renal disorders [1]. CKD is more common in low- and middle-income countries than in high-income countries [2]. CKD is most commonly attributed to DM and hypertension, but other causes, such as glomerulonephritis, infection, and environmental exposure (air pollution and pesticides), are common in many developing countries [3]. Genetic risk factors can also contribute to the onset of CKD [4]. If CKD is diagnosed in the early stages (1, 2, 3), it can alter the severity of early CKD and reduce complications. In the early stages of CKD, symptoms are usually not noticeable, with a significant reduction in kidney function being the first sign of disease [5]. Extensive kidney damage is seen in stages 4 and 5, usually resulting in ESRD [5]. DM is the leading cause of CKD; about 40% of patients with DM develop CKD [6]. The presence of microalbuminuria is well established in patients with type 1 diabetes (DM1). Microalbuminuria eventually progressed to macroalbuminuria and decreased GFR. However, patients with type 2 DM (T2DM) present heterogeneous clinical features. Groups of patients with T2DM who have decreased GFR with normal albuminuria have been identified. T2DM produces metabolic changes that alter renal hemodynamics by promoting glomerular hyperfiltration and glomerular enlargement [7]. Even though 40% of patients with T2DM develop glomerular hyperfiltration, the mechanisms responsible are still unclear [8]

Other factors, such as oxidative stress and low-grade inflammation, may be linked to the pathological development of CKD in T2DM [9,10].

Currently, the diagnosis of CKD is made by blood urea and serum creatinine (sCr) levels; however, sCr concentration is not very reliable due to its dependence on the patient’s body weight, age, and gender [11]. However, GFR is considered the gold standard for evaluating kidney function [12].

Low-grade inflammation is considered a hallmark of CKD and is involved in developing all-cause mortality, even in the early stages [13].

Oxidative stress (OS) is defined as the imbalance between reactive oxygen species (ROSs), reactive nitrogen species (RNS), and free radicals as oxidants [14]. Oxidants are unstable molecules that promote oxidation reactions with other molecules, such as proteins, lipids, and DNA, to stabilize themselves. ROSs are characterized by an unpaired electron on an oxygen atom. ROSs include free radicals, hydroperoxides (HO2•), the superoxide anion radical (O2•−), and the hydroxyl radical (HO•). Redox signaling agents have oxidizing properties such as hydrogen peroxide (H_2_O_2_) or singlet oxygen (1O_3_). OS reactions also involve oxygen compounds, such as non-radical compounds (peroxynitrite (ONOO-) and hypochlorous acid (HOCl) [15]. The metabolic imbalance of OS leads to the depletion of endogenous and exogenous antioxidants, abnormal protein function, damage to deoxyribonucleic acid (DNA) and enzymes, and finally, irreversible cell damage [16]. In diabetic CKD, the production of ROS in the kidney is mainly due to the increased expression and activity of NADPH oxidase (NOX) [17]. Increased NOX-mediated ROSs in the kidney is considered a mediator of glomerulosclerosis, endothelial dysfunction, the development of albuminuria, and disruption of glomerular hemodynamics [10]

This study aimed to evaluate inflammation and OS state in the early stages of CKD in patients with and without T2DM.

## 2. Results

Table 1 shows the consecutive decrease in Hb levels as kidney disease progressed in patients without DM (*p* = 0.04), with T2DM (*p* < 0.01), and between those without DM and with T2DM (*p* < 0.01). A similar decrease was observed in hematocrit, with a significant difference in patients with DM (*p* < 0.01), even between patients with and without DM (*p* < 0.01). As expected, glucose was found to be increased in patients with T2DM (*p* < 0.01). sCr levels increased as the disease stage progressed in patients with and without DM (*p* < 0.01). Low-density cholesterol and very-low-density cholesterol showed altered levels in patients with DM compared to patients without DM (*p* = 0.04). Patients with and without DM in the three stages did not show a trend toward decreasing or increasing vitamin D. FGF23 levels decreased as the CKD stage progressed in T2DM patients. Patients in stage 3 with T2DM showed significantly lower levels of FGF-23 (*p* = 0.05).

Parathyroid hormone levels increased as kidney disease progressed in patients without DM (*p* = 0.03) and were higher in patients with T2DM. A significant difference was found between patients without DM and an increase in parathyroid hormone in patients with T2DM (*p <* 0.01). GFR decreased as the stage of kidney disease progressed, both in patients with and without DM (*p* < 0.01).

The levels of LPO were similar in stages 1, 2, and 3 of patients with and without DM. 

NO levels in the group of patients with T2DM, 2470.63 ± 422.83 µg/mL, were found to be significantly increased compared to the levels of the group of patients without DM, 1921.99 ± 1328.03 µg/mL (*p* < 0.01). However, NO levels in patients without DM decreased as the early stage of kidney disease progressed (*p* < 0.01) (Table 2 and Table 3).

The activity of the antioxidant enzyme SOD in the group of patients with T2DM, 14.64 ± 6.10 U/L, was found to be significantly decreased (*p* < 0.01) compared to the activity in the group of patients without DM, 17.24 ± 7.02 U/L. SOD enzyme activity increased as the stage of kidney disease progressed in patients without DM (*p* = 0.04) and with T2DM (Table 2 and Table 3).

The activity of the antioxidant enzyme GPx was found to be significantly increased in the group of patients with T2DM, 10.86 ± 0.30 nmol/min/mL (*p* < 0.01), compared to the group of patients without DM, 10.79 ± 0.38 nmol/min/mL. However, the activity of the antioxidant enzyme in the early stages of kidney disease in patients with and without DM did not show statistical significance (Table 2 and Table 3).

The total antioxidant capacity showed similar results between the early stages of kidney disease in patients with and without DM. However, patients in the group with T2DM showed significantly lower levels, 1.24 ± 0.44 µM, than those in the group without DM, 1.57 ± 0.63 µM (*p* < 0.01). In the early stages of kidney disease, no significant differences were observed in patients with and without DM (Table 2 and Table 3).

The expression of the pro-inflammatory cytokines IL-6 was found to be significantly increased in the group of patients with T2DM, 57.23 ± 23.37 pg/mL (*p* < 0.01) compared to the group of patients without DM, 45.81 ± 11.25 pg/mL. However, in the early stages of kidney disease, no significant differences were observed in both patients with and without DM (Table 2 and Table 3).

The expression of TNF-α showed similar behavior with significantly higher expression in the group of patients with T2DM, 6932.09 pg/mL (*p* < 0.01), compared to patients without DM, 6.31 ± 2.01 pg/mL. The analysis of TNF-α in the early stages of kidney disease in the groups of patients with and without DM did not show significant differences (Table 2 and Table 3).

## 3. Discussion

The definition of CKD is a broader concept than simply chronic renal failure because it encompasses patients who may have relatively preserved renal function. In the early stages of kidney disease, patients may be at risk, as there may be increased severity of the disease, which increases the risk of CVD. A decreased GFR (<60 mL/min/1.73 m^2^) is by itself a diagnostic parameter of CKD when it persists for a period ≥ 3 months [18]. The evolution of CKD is usually slow, with few characteristic symptoms that show the clinical picture, which means that the diagnosis is usually not made until there is renal failure [19]. Therefore, the present study evaluated the impact of inflammation and OS markers in the early stages of CKD in patients with and without T2DM. 

The kidneys play a central role in potassium (K) homeostasis, CKD being an especially prominent risk factor for the development of hyperkalemia [20]. The ability of the kidneys to excrete K is inversely related to GFR [21]. In patients with CKD, the adaptive increase in renal and gastrointestinal K excretion helps prevent hyperkalemia while maintaining a GFR ≥15-20 mL/min/1.73 m^2^ [22]. Some observational studies have suggested that the risks associated with hyperkalemia are more significant in individuals with normal renal function than those with CKD due to the progressive increase in K [23,24]. In the present study, serum K levels were significantly increased in patients without DM, suggesting that renal deterioration is advancing.

sCr is a product of muscle metabolism, which, like urea, is excreted by the kidneys. The sCr level is relatively stable and is therefore considered a reliable indicator of renal function [25]. sCr levels increased according to the stage of the disease in the included patients. sCr levels were inversely proportional to the decrease in GFR [26]. Although it may be helpful to determine sCr when kidney disease is suspected, validated epidemiological studies do not differentiate the increase in sCr by etiology [27]. In the present study, sCr levels were measured in addition to GFR and other metabolites.

GFR is considered the gold standard for the evaluation of renal function. GFR is relatively low at birth but increases during childhood to approximately 120 mL/min/1.73 m^2^ in adulthood [28]. In the present study, decreased GFR was observed as the early stage of CKD progressed in patients with and without T2DM. As previously reported, our findings show that higher OS markers are associated with greater CKD severity [29]. 

FGF-23 is a phosphaturic hormone that is elevated in the later stages of kidney disease, followed by a reduction in 1,25-dihydroxy-vitamin D (1,25 [OH]2D) and an increase in parathyroid hormone (PTH) [30]. It was recently published that the administration of an intermittent regimen of PTH administered in the early stages of CKD could prevent the increase in FGF-23 through its phosphaturic action. It is noteworthy that patients with T2DM in the early stages of CKD had a significant decrease in FGF-23, contrary to what was previously reported, where 100% of patients with CKD in stage 5 had elevated levels of FGF-23 and PTH [30,31]. The previous results suggest that beginning in the early stages of CKD, there are changes identifiable by diagnostic methods of FGF-23 and PTH as GFR falls below 60 mL/min/1.73 m^2^, and phosphate is retained. PTH synthesis and secretion are stimulated [32,33]. In the present study, the obtained concentrations of PTH increased as the stage of kidney disease progressed in patients with and without DM. The results found in the present study of patients in the early stages of CKD suggest that the greater regulation of PTH secretion could respond to FGF23 through a pathway driven by Na+/K+-ATPase [34].

As GFR falls below 60 mL/min/1.73 m^2^, phosphate is retained, and PTH synthesis and secretion are stimulated [32,33]. In the present study, the obtained concentrations of PTH increased as the stage of kidney disease progressed in patients with and without DM. Due to the increase in OS in CKD, a proportion of PTH is oxidized in patients with kidney disease, which means that the hormone is not biologically active [35]. It has even been reported that PTH affects endothelial function through ROS production in the mitochondria, altering various signaling pathways [36]. In experimental studies, it has been determined that high hormone concentrations can affect the expression of the enzyme NO synthase because patients with more advanced stages of kidney disease present an inversely proportional relationship between the concentrations of NO and PTH [37]. 

Vitamin D metabolism is disrupted by the failure of the second hydroxylation step of 25-hydroxyvitamin D to occur, converting it to the active form 1,25-dihydroxy vitamin D produced in the kidney [38]. Inhibition of 1,25-dihydroxy vitamin D induces hypocalcemia, stimulating the parathyroid gland to release PTH at persistent circulating levels [39]. 

Malnourished patients have elevated levels of C-reactive protein (CRP), IL-6, TNF-α, and concomitant CVD when they reach the end stage. Many diseases that cause CKD, such as DM and hypertension, are associated with CVD. The direct effect of renal failure per se that directly contributes to the inflammation–malnutrition–atherosclerosis paradigm is not fully established in the early stages of CKD [40,41]. 

The imbalance between oxidants and the reduction in antioxidant defenses favor kidney damage, limit cell repair mechanisms, and accelerate functional deterioration. In several cross-sectional studies, plasma levels of OS and markers of inflammation were found to be associated with advanced stages of CKD. However, this did not correlate with decreased GFR [42]. In longitudinal studies, the same behavior between GFR and OS has been described, suggesting that changes may influence the concentration of these biomarkers in glomerular and tubular function [43].

In the present study, Hb decreased as the stage of kidney disease progressed. Various factors could explain this; we must consider that erythrocytes play a fundamental role in maintaining the oxidative status of the blood. There are several mechanisms by which oxidative damage and inflammation cause a decrease in Hb by iron deficiency in patients with CKD, including a reduction in antioxidants and oxidation of erythrocyte membrane phospholipids [44]. Other mechanisms are the absolute deficiency of iron characterized by reduced or absent stores. Functional iron deficiency is another mechanism where iron’s availability to be incorporated into erythroid precursors is insufficient due to increased hepcidin levels [45].

Furthermore, CKD predisposes patients to vitamin and mineral deficiencies, which contribute to anemia, CVD, and metabolic imbalances [46]. There are longitudinal studies that show that as CKD progresses, Hb levels decrease until anemia occurs [47]. Oxidative damage causes premature red blood cell aging by the increased production of hepcidin, reduced production of the iron transport protein transferrin and its receptor, and induction of resistance to erythropoietin [47]. Therefore, the combination of factors may be responsible for the decrease in Hb in patients with early stages of CKD included in the study

NO is a molecule with a wide variety of physiological functions, including; maintaining muscle tone and functions as an intracellular messenger and cytotoxic agent. NO has a substantial effect on the vascular endothelium through its bioavailability. When NO levels increase significantly, it is considered an RNS [48]. In the present study, significantly increased levels of NO were found in the group of patients with T2DM, which suggests that NO functions as an oxidant. On the other hand, NO levels decreased as the early stages of kidney disease progressed in patients without DM, affecting the bioavailability of NO in the vascular endothelium. In experimental studies, a decrease in NO has been reported, which alters the correct function of the endothelium; favors cardiac hypertrophy, fibrosis, and OS; and increases levels of natriuretic peptides as kidney disease progresses [49,50]. NO works with mechanisms that underlie concepts that have arisen. One mechanism is the ability of Hb to combine with NO and play a vital physiological role characterized by hypoxic vasodilation. Hypoxic vasodilation involves the release of vasodilator substances without direct dependence on the presence of endothelial NO synthase [51,52]. In the present study, we found a decrease in Hb and an increase in NO as the CKD stage progressed. These data could suggest the combination of factors, hypoxic vasodilation, and decreased endothelial bioavailability of NO. 

The SOD enzyme is the body’s first antioxidant defense [53]. The activity of the SOD enzyme presented a significant and proportional increase according to the evolution of the early stage of kidney disease. The analysis of the groups of patients with and without T2DM found a substantial decrease in SOD enzyme activity in patients with T2DM, which could suggest early injury to glomerular mesangial cells. Glomerular mesangial cell injury has been previously identified as an early risk factor in the development of diabetic nephropathy [54].

The GPx enzyme uses glutathione to reduce H_2_O_2_ to organic peroxides by acting in conjunction with the enzyme peroxynitrite reductase [55]. In the present study, the activity of the GPx enzyme was similar in the three early stages of kidney disease in patients with and without DM, suggesting that the GPx enzyme’s action was preserved even though GFR declined as the disease progressed. However, the group of patients with DM showed a significant increase compared to those without DM. Endogenous antioxidant enzyme defenses can be modified with changes in oxidative stress markers, as in the present study. It was previously reported that as GFR decreases, GPx enzyme activity decreases [56]. The results on the effect of antioxidants in human diabetic nephropathy are limited [57].

TAC measures the global effect of antioxidant mechanisms on serum oxidative *status*. The high antioxidant activity reflects protection or benefits in response to chronic OS [58]. In this study, serum TAC concentrations did not show significant changes as the early stages of kidney disease progressed. The determination of TAC in the groups showed a substantial decrease in the marker in patients with T2DM included in the study compared to patients without DM. However, the underlying mechanisms of the renoprotective effect of the TAC marker were not elucidated [59,60].

The expression of the pro-inflammatory cytokines TNF-α and IL-6 was similar between patients with and without DM in the early stages of kidney disease. However, when the groups with and without DM were analyzed, a significant increase in pro-inflammatory cytokines IL-6 and TNF-α was observed in the T2DM group, suggesting that DM plays an essential role in the CKD evolution compared to patients without DM. It is important to consider that diabetic nephropathy is the most common complication of T2DM and is the leading cause of ESRD worldwide [61].

Arterial hypertension, DM, dyslipidemia, and microalbuminuria are essential predictors of identifying people at risk of kidney disease [62]. In the present study, we included 71 patients with early kidney disease without DM and 65 with T2DM who had hypertension. In this regard, it was previously reported that in the early stages of diabetic kidney disease, when renal function is still well-preserved, systemic blood pressure is already elevated. The same phenomenon also occurs in patients without DM [62]. There is little information available regarding the state of inflammation and oxidative stress in the early stages of early kidney disease in patients with and without DM

The kidneys play an important role in maintaining homeostasis in the body. Early recognition and intervention are essential to delaying disease severity, maintaining quality of life, and improving survival. Primary care physicians have the opportunity and responsibility to assess at-risk patients, identify affected patients, and facilitate the impact of early stage kidney disease by initiating early therapy and monitoring disease severity. The basic recommendations consist of intensive blood pressure control, with a goal of ≤130/80 mmHg. Angiotensin-converting enzyme inhibitors and angiotensin-II receptor antagonists are the most effective due to their unique ability to decrease proteinuria. Hyperglycemia with an AIC concentration <7% should be recommended. Anemia should be maintained at hemoglobin levels of 11-12 g per dL (110-120 g per L). Dietary restrictions on Pi are imperative to prevent hyperparathyroidism disease as much as possible, in addition to using antacids and vitamin D supplements. 

In conclusion, oxidative imbalance was determined in patients with and without T2DM. The presence of OS in T2DM was characterized by an increase in oxidative markers and a decrease in antioxidant defenses (SOD, TAC, and GPx). Only patients without T2DM showed a significant increase in the activity of the antioxidant enzyme SOD and a substantial decrease in NO when the CKD stage increased. In the present study, SOD and NO markers changed during the early stages of CDK, similarly to those of the clinical parameters Hb, hematocrit, and sCr frequently used to assess CDK progression. NO and SOD markers could help detect the early stages of CKD in patients without T2DM. Detecting and treating early kidney disease offers an invaluable opportunity to use the different actions available to delay the progression of the disease and prevent its evolution to ESRD.

## 4. Patients and Methods

An analytical cross-sectional study was carried out on patients diagnosed with CKD in early stages (1, 2, 3) with and without T2DM treated at the Department of Nephrology of the Civil Hospital of Guadalajara “Juan I. Menchaca” in Guadalajara, Jalisco, Mexico, who agreed to participate in the study and signed informed consent. Male and female patients aged 18–60 years were included. Patients who ingested antioxidants (vitamin E, vitamin C, etc.) within the three months before the study, patients with clinical or biochemical data of an infectious process, and patients who were undergoing renal replacement therapy (RRT) were not included. Anthropometric, clinical, and biochemical data were measured: body mass index (BMI Kg/m^2^), systolic blood pressure (SBP) and diastolic blood pressure (DBP), hemoglobin (Hb), hematocrit, glucose, sCr, total cholesterol, high-density cholesterol, low-density cholesterol, very-low-density cholesterol, triglycerides, albumin, sodium, potassium, chlorine, calcium, phosphorus, magnesium, FGF-23, vitamin D, parathyroid hormone, 24 h urine protein, and GFR.

Eighty-eight patients were without DM and one hundred with T2DM. There were eighty-one men and one hundred and seven women. The average age was 50.1 ± 22.02 years. Table 4 shows that 71 patients without DM had hypertension, and 65 patients with T2DM had hypertension. Alcohol consumption was reported in 13 patients without DM and 7 with T2DM. A total of 38 patients without DM were smokers, as opposed to 55 with T2DM. Eight patients without DM were in stage 1, 34 in stage 2, and 46 in stage 3. Of the patients with T2DM, 10 were in stage 1, 37 in stage 2, and 53 in stage 3. 

### 4.1. Blood Samples 

A total of 5 mL of venous blood was obtained in a test tube containing ethylenediaminetetraacetic acid (EDTA) for plasma and 5 mL in a dry tube for serum. Plasma and serum were extracted at 1800 revolutions per min (rpm) for × 10 min. The samples were then stored at −80 °C until final processing.

### 4.2. Inflammation Markers

#### IL-6 and TNF-α

The commercial kit 900-K25 from the manufacturer Peprotech^®^ Inc. was used. The method is a sandwich ELISA assay. Absorbance at 405 nm with 650 nm correction was recorded. The standard curve was prepared simultaneously to interpolate the concentrations of each sample.

### 4.3. Oxidants

#### 4.3.1. Nitric Oxide

Before determining the NO levels, the serum samples were deproteinized by adding six mg of zinc sulfate to four hundred μL of the example, vortexed for one min, and the samples were centrifuged at 10,000× *g* for ten min at 4 °C. The colorimetric method was used according to the kit (NO assay kit, User protocol 482650, Calbiochem^®^, USA) to determine NO. The plate was read at 540 nm in a spectrophotometer within the first twenty min of completion of the procedure.

#### 4.3.2. Lipoperoxides 

According to the manufacturer’s instructions, the levels of LPO in plasma were measured through the FR22 assay kit (Oxford Biomedical Research Inc., Oxford, MI, USA^®^). The limit of detection for this test was 0.1 nmol/mL. The chromogenic reagent reacts with malondialdehyde (MDA) and 4-hydroxy-alkenes to form a stable chromophore. The pattern curve with known concentrations of 1,1,3,3-tetra methoxy propane in Tris-HCl was used. The intra-assay CV was 8.5%.

### 4.4. Antioxidants 

#### 4.4.1. Superoxide Dismutase

We followed the kit manufacturer’s instructions (SOD No. 706002, Cayman Chemical Company^®^, Ann Arbor, MI, USA). The reaction of tetrazolium salts made it possible to detect the O2- generated by the enzymes xanthine oxidase and hypoxanthine. The serum samples were diluted 1:2 in the sample buffer. The absorbency was read at 440 wavelengths of nm. The levels are reported in IU/mL.

#### 4.4.2. Glutathione Peroxidase

Glutathione peroxidase was determined according to the commercial kit 703102 from the manufacturer Cayman chemical Company^®^, Ann Arbor, MI, USA. This method measures the decrease in absorbance caused by the reaction of an organic peroxide (tertbutyl hydroperoxide) with GP× of the sample studied in the presence of NADPH. The absorbance obtained at 340 nm was recorded for 30, 60, 90, 120, 150, and 180 s. Subsequently, the rate of decrease in absorbance per min related to the concentration of NADPH consumed in the reaction was determined.

#### 4.4.3. Total Antioxidant Capacity

The evaluations of TAC were made following the instructions of the kit manufacturer (Total Antioxidant Power Kit, No. TA02.090130, Oxford Biomedical Research^®^, MI, USA) to obtain the concentration in mM equivalents of uric acid. The detection limit was 0.075 mM. The dilution factor was considered in the result. The intra-assay CV was 7.8%.

#### 4.4.4. Human Fibroblast Growth Factor 23

To determine FGF23, protein concentration was analyzed by an FGF23 ELISA (MBS263043, MyBioSource^®^, San Diego, CA, USA). The method was followed as suggested by the manufacturer of the ELISA kit. This experiment uses the double-sandwich ELISA technique. The optical density was read at 450 nm. The Synergy HT (BIOTEK) microplate reader was used for all of the technical readings of optical density.

#### 4.4.5. Parathyroid Hormone Intact (PTH)

The PTH concentrations were determined through the use of a commercial kit (21-IPTHU-E01, ALPCO^®^, Salem, MA, USA). The manufacturer’s instructions were followed. This ELISA detects only the protein’s intact, biologically active portion. The optical density was read at 450 nm. 

#### 4.4.6. 25 (OH)-Vitamin D Direct

The kit manufacturer’s instructions were followed (25 (OH)—vitamin D direct Elisa Kit No. K2109KO, ALPCO ^®^, Salem, MA, USA). This assay utilizes a competitive ELISA technique. The optical density was read at 450 nm. 

### 4.5. Statistical Analysis

Quantitative variables are shown as mean ± standard deviation (SD). Nominal variables in numbers. The Kolmogorov–Smirnov test was performed to determine the data distribution of the entire sample. Comparisons were made using the Mann–Whitney U test. For multiple differences between groups, the Kruskal–Wallis test was used, followed by Dunn’s test with Bonferroni adjustment at *p* ≤ 0.05 (Dunn-Bonferroni). Any value of *p* ≤ 0.05 was considered significant. All tests were performed with IBM SPSS statistics version 23.

### 4.6. Ethical Considerations

This study adhered to the ethical principles for medical research on human beings stipulated in the Declaration of Helsinki 64th General Assembly, Fortaleza, Brazil, October 2013, in the Belmont Report and Standards of Good Clinical Practices according to the guidelines of the International Conference on Harmonization by the provisions of the General Health Law by the Regulations of the General Health Law on Research for Health art. 17. This study corresponds to category II (research with minimal risk). It required an Informed Consent Letter. This study was authorized by the Research and Ethics Committee of the Civil Hospital of Guadalajara “Juan I. Menchaca” with the State Registration Number 069/15 HCJM/2015. 

## Figures and Tables

**Table 1 ijms-23-11196-t001:** Clinical and biochemical data.

	No DM				DM				
	Stage 1 N = 8	Stage 2 N = 34	Stage 3 N = 46	*K-W* *p*	Stage 1 N = 10	Stage 2 N = 37	Stage 3 N = 53	*K-W* *p*	*M-W* *p*
Body Mass Index (Kg/m^2^)	25.64 ± 4.1	28.97 ± 5.82	28.17 ± 6.72	0.4	29.9 ± 5.1	29.2 ± 5.4	28.94 ± 7.19	0.7	0.3
SBP (mmHg)	130 ± 8.4	126.1 ± 21.9	129 ± 23	0.8	128.5 ± 17.7	120.5 ± 36.6	129.36 ±3 1.42	0.6	0.6
DBP (mmHg)	78 ± 13.3	76.2 ± 12.6	82 ± 16	0.2	73.9 ± 3.2	72.8 ± 22.3	75.96 ± 18.5	0.8	0.4
Hemoglobin (mg/dL)	14.61 ± 0.61	14 ± 1.4	13.02 ± 2.13	0.04 *	13.9 ± 1.2	13.48 ± 2.1	12.39 ± 1.69	<0.01 **	<0.01 **
Hematocrit (%)	43.17 ± 1.77	41 ± 8.2	40 ± 6	0.1	41.6 ± 3.7	39.9 ± 6.5	37.26 ± 4.98	<0.01 **	<0.01 **
Glucose (mg/dL)	90.5 ± 3.1	94.02 ± 16.1	93.4 ± 23.4	0.8	149.5 ± 62	158.86 ± 62.6	144.18 ± 70.46	0.3	<0.01 **
Creatinine (mg/dL)	0.77 ± 0.07	1.02 ± 0.19 ^a,b^	1.65 ± 0.4	<0.01 *	0.7 ± 0.15	0.96 ± 0.17	1.43 ± 0.31	<0.01 **	0.04 **
Total cholesterol (mg/dL)	173.4 ± 31	186 ± 55.1	179.3 ± 36.1	0.9	169.6 ± 83.3	177.5 ± 39.9	176.55 ± 47.09	0.2	0.2
HDL (mg/dL)	38.7 ± 2.2	41 ± 11.7	44.6 ± 14.2	0.6	47.4 ± 15.2	38.6 ± 8.6	42.08 ± 11.21	0.2	0.8
LDL (mg/dL)	35.3 ± 21	31.3 ± 14.2	30.4 ± 12.4	0.9	37.5 ± 25.04	36.4 ± 15.3	39.09 ± 22.35	0.9	0.04 **
VLDL	97.6 ± 28.7	109.5 ± 49.2	103.1 ± 27.4	0.8	86.3 ± 53.1	103.3 ± 37.6	91.98 ± 36.9	0.2	0.04 **
Triglycerides (mg/dL)	158 ± 100.4	161.03 ± 76	153 ± 65.14	0.9	188.1 ± 41.6	178.52 ± 13.21	193.08 ± 15.45	0.9	0.051
Albumin (mg/dL)	2.95 ± 2.3	3.6 ± 0.45	3.6 ± 0.62	0.9	2.6 ± 0.94	3.9 ± 0.43	3.76 ± 1.1	0.3	0.7
Sodium (mmol/L)	138.4 ± 5.9	138.3 ± 7.7	138 ± 4.7	0.2	138.1 ± 2.8	136.03 ± 4.8	136.88 ± 5.21	0.7	<0.01 **
Potassium (mmol/L)	4.01 ± 0.28 ^b^	4.4 ± 0.63	4.6 ± 0.52	<0.01 *	4.23 ± 0.61	4.52 ± 0.67	4.71 ± 0.83	0.5	0.4
Chlorine (mmol/L)	105.3 ± 5.02	107.3 ± 4.12	106 ± 5.5	0.6	105.2 ± 3.8	102.2 ± 6.2	104.56 ± 6.01	0.3	<0.01 **
Calcium (mmol/L)	8.9 ± 1.2	8.9 ± 1.3	8.9 ± 0.52	0.3	9.2 ± 1.3	9.1 ± 0.52	8.92 ± 0.53	0.3	0.9
Phosphorus (mmol/L)	3.9 ± 0.52	3.6 ± 0.9	3.6 ± 0.7	0.5	3.5 ± 0.31	3.42 ± 0.67	3.68 ± 0.62	0.3	0.8
Magnesium (mmol/L)	1.9 ± 0.6	2.04 ± 0.61	1.9 ± 0.3	0.4	1.8 ± 0.16	1.83 ± 0.24	1.92 ± 0.33	0.3	0.3
FGF 23 (ng/mL)	97.1 ± 80.8	169.3 ± 82.2	89.04 ± 26.5	0.1	269.78 ± 110.9	118.14 ± 59.68	103.59 ± 22.83	0.05	0.7
Vitamin D (nmol/L)	52.5 ± 31.9	48 ± 25	46.7 ± 29.1	0.6	26.8 ± 19.8	32.28 ± 4.39	26.89 ± 18.97	0.6	<0.01 **
Parathyroid hormone (pg/mL)	31.6 ± 17.5	35.4 ± 26.3	56.9 ± 8.7	0.03 *	56 ± 13.51	59.9 ± 5.9	76.76 ± 6.41	0.1	<0.01 **
GFR (mL/min/1.73 m^2^)	115.1 ± 14.4	74.3 ± 8.7 ^a,b^	41.8 ± 8.5	<0.01 *	100.4 ± 10.9	74.92 ± 8.45	45.45 ± 8.9	<0.01 **	0.7
24 h urine protein	116.1 ± 62	73 ± 26.3	97.6 ± 25.4	0.9	154.7 ± 145.2	82.5 ± 69.85	103.66 ± 61.54	0.7	0.4

The results are expressed as mean ± standard deviation. GFR: glomerular filtration rate; SBP: systolic blood pressure; DBP: diastolic blood pressure. * *K-W*: Kruskal–Wallis test. ** *M-W*: Mann–Whitney U test comparing the median values between DM and T2DM. a and b indicate significant difference compared to stages 1 and 2, respectively, through the Dunn–Bonferroni post hoc test.

**Table 2 ijms-23-11196-t002:** Inflammatory and oxidative stress markers.

	No DM				DM			
	Stage 1 N = 8	Stage 2 N = 34	Stage 3 N = 46	*K-W* *p*	Stage 1 N = 8	Stage 2 N = 34	Stage 3 N = 46	*K-W* *p*
**Oxidants**								
Lipoperoxides (mM)	46.7 ± 44	24.6 ± 12.03	30.2 ± 23	0.5	35.44 ± 18.32	24.24 ± 3.06	26.54 ± 16.68	0.1
Nitric oxide (µg/mL)	2337.5 ± 1979.7 ^c^	2230.9 ± 1680.9	1591.7 ± 630.6 ^a^	<0.01 *	2300.43 ± 1279.29	2258.58 ± 566.12	2658.18 ± 575.11	0.9
**Antioxidants**								
Superoxide dismutase (U/L)	14.5 ± 3.9	15.1 ± 4.1 ^c^	19.4 ± 8.5 ^b^	0.04 *	12.71 ± 3.74	13.4 ± 4.64	15.83 ± 7.06	0.1
Glutathione peroxidase (nmol/min/mL)	10.62 ± 0.21	10.8 ± 2.28	10.8 ± 0.5	0.1	10.77 ± 0.13	10.87 ± 0.35	10.87 ± 0.28	0.5
Total Antioxidant Capacity (µM)	1.64 ± 0.51	1.6 ± 0.64	1.53 ± 0.7	0.7	1.42 ± 0.54	1.24 ± 0.39	1.21 ± 0.46	0.4
**Pro-inflammatory cytokines**								
Interleukine 6 (pg/mL)	47.5 ± 12.02	46.32 ± 12.5	45.1 ± 10.2	0.9	60.32 ± 32.75	55.12 ± 23.32	58.09 ± 21.73	0.8
Tumor Necrosis Factor α (pg/mL)	6.32 ± 0.93	6.3 ± 1.6	6.3 ± 2.5	0.5	8.16 ± 2.6	6.62 ± 1.49	6.92 ± 2.3	0.3

The results are expressed as mean ± standard deviation. * *K-W*: Kruskal–Wallis test. a, b, and c indicates significant difference compared to stages 1, 2, and 3, respectively, through the Dunn–Bonferroni post hoc test.

**Table 3 ijms-23-11196-t003:** Inflammatory and oxidative stress markers of patients with and without DM.

	No DM	DM	*p*
**Oxidants**			
Lipoperoxides (mM)	29.3 ± 22.40	26.58 ± 24.25	0.1
Nitric oxide (µg/mL)	1921.99 ± 1328.03	2470.63 ± 422.83	<0.01 *
**Antioxidants**			
Superoxide dismutase (U/L)	17.24 ± 7.02	14.64 ± 6.10	<0.01 *
Glutathione peroxidase (nmol/min/mL)	10.79 ± 0.38	10.86 ± 0.30	<0.01 *
Total Antioxidant Capacity (µM)	1.57 ± 0.63	1.24 ± 0.44	<0.01 *
**Pro-inflammatory cytokines**			
Interleukine 6 (pg/mL)	45.81 ± 11.25	57.23 ± 23.37	<0.01 *
Tumor Necrosis Factor α (pg/mL)	6.31 ± 2.01	6.932.09	<0.01 *

The results are expressed as mean ± standard deviation. * M-W: Mann–Whitney U test.

**Table 4 ijms-23-11196-t004:** Demographic data.

	No DM	T2DM	*p*
**Age**	50 ± 22	62 ± 14	<0.01 *
**Gender**			
Male (n)	39	42	0.431
Female (n)	49	58	
**Stage**			
1 (n)	8	10	0.962
2 (n)	34	37	
3 (n)	46	53	
**Hypertension**			
Hypertensive (n)	71	65	<0.01 **
Not hypertensive (n)	17	35	
**Alcohol**			
Alcohol consumption (n)	13	7	0.068
No alcohol consumption (n)	75	93	
**Tobacco consumption**			
Smoker (n)	38	55	0.917
Nonsmoker (n)	50	45	

The results are expressed in numbers and mean ± standard deviation. DM: diabetes mellitus. * Significant for the *t*-Student test. ** Significant for Chi^2^ test.

## Data Availability

We guarantee the integrity of the data shown here. If the reviewers require the database, we can provide each of the documents with prior authorization from the Ethics and Research Committee.

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
