# Peer review of "The Influence of the Severity of Early Chronic Kidney Disease on Oxidative Stress in Patients with and without Type 2 Diabetes Mellitus"

_ijms, 2022, doi:10.3390/ijms231911196_

Round 1

Reviewer 1 Report

Here, the authors report the level of various measures of oxidative stress in the context of CKD in patients with out without diabetes mellitus. Unsurprisingly, markers of renal failure are associated with the stage of renal failure.  There is some evidence that markers of oxidative stress correlate with the CKD staging, and that DM may have exacerbated some of these effects. In general, the organization of the manuscript is poor and the discussion mostly rehashes the results of the study with little insight regarding the significance (or lack thereof) of individual findings. Consequently, the “take home message” is unclear.  Specifically, does DM contribute to functional declines associated with CKD as a consequence of increased oxidative burden?

Some specific comments:

There are some grammatical issues and/or typos throughout the ms.

The reported age range is rather large (18 – 60), hence age itself is likely to be a contributing factor to some of the observed effects.  Was age built into the model?

If samples were collected on EDTA, there should not be blood serum present.

There is no need to provide a step-for-step description of the methods used for commercially available assay kits.  

The description of the statistical models used needs to be more clear. Why was a non-parametric test used for the “ANOVA” but a parametric test used for the post-hoc? Was the Kolmogorov used within each condition (DM versus non-DM only) across stages of CKD? Also, how was the Mann-Whitney U-test applied? There is a single p-value, but this is equivalent to a t-test. Which significant difference(s) is being highlighted?

The finding that glucose was significantly elevated in patients with DM is what defines DM in the first place. Why is this reported? To confirm the diagnosis? Why wasn’t serum insulin level also reported? Did all the patients have Type I and/or Type II diabetes? Does this matter?

The use of the lowercase letters in the tables as indicators of significance is unclear. Specifically, what is different from what? For the creatinine level, there is an ‘a’ and a ‘b’ for the Stage 2 value in the No DM group, so does this mean it is different from both the Stage 1 and Stage 3 patients?   

Why does the numbering scheme for citations switch to roman numerals? 

Author Response

Reviewer 1

Comment. Here, the authors report the level of various measures of oxidative stress in the context of CKD in patients with out without diabetes mellitus. Unsurprisingly, markers of renal failure are associated with the stage of renal failure.  There is some evidence that markers of oxidative stress correlate with the CKD staging, and that DM may have exacerbated some of these effects. In general, the organization of the manuscript is poor and the discussion mostly rehashes the results of the study with little insight regarding the significance (or lack thereof) of individual findings. Consequently, the “take home message” is unclear.  Specifically, does DM contribute to functional declines associated with CKD as a consequence of increased oxidative burden?

Answer. Thanks for the comment; you are right. We restructured the document with a complete statistical analysis. What we can take home from the investigation lies in the conclusion

“Patients with T2DM present higher levels of oxidative and inflammatory markers accompanied by a decrease in antioxidant defense. However, these oxidative status markers were associated with CKD stage progression in patients without T2DM. NO and SOD markers could help detect the early stages of CKD in patients who have not yet developed metabolic comorbidities such as T2DM.”

Some specific comments:

Comment. There are some grammatical issues and/or typos throughout the ms.

Answer. I apologize for the grammatical and spelling errors, the document has been restructured, and the grammar checked. We hope that the mistakes have decreased considerably

Comment. The reported age range is rather extensive (18–60); hence age itself is likely to be a contributing factor to some of the observed effects.  Was age built into the model?

Answer. Thank you for the question; the age of the patients is included in Table 1, indeed the age of the included patients could be a confounding factor

Comment. If samples were collected on EDTA, there should not be blood serum present.

Answer. Five mL of venous blood was obtained in a test tube containing ethylenediaminetetraacetic acid (EDTA) for plasma and 5 mL in a dry tube for serum.

Comment. There is no need to provide a step-for-step description of the methods used for commercially available assay kits.  

Answer. Significantly shortened the wording of the step-by-step methods. We hope that the current description of the methods has been sufficiently clear

Comment. The description of the statistical models used needs to be clearer. Why was a non-parametric test used for the “ANOVA” but a parametric test used for the post-hoc? Was the Kolmogorov used within each condition (DM versus non-DM only) across stages of CKD? Also, how was the Mann-Whitney U-test applied? There is a single p-value, but this is equivalent to a t-test. Which significant difference(s) is being highlighted?

Answer. Kolmogorov's test was used to determine the data distribution for the entire sample. The Mann-Whitney U-test in Tables 2 and 3 was used to compare the data obtained between the groups DM vs. non-DM, but these values were not shown in the tables. Therefore, the results of the Mann-Whitney U-test from Table 3 are removed and moved to a new Table that compares DM and non-DM. The statistical analysis section is corrected as follows:

“Quantitative variables are shown as mean ± standard deviation (SD). Nominal variables in numbers. The Kolmogorov-Smirnov test was performed to determine the data distribution of the entire sample. Comparisons were made using the Mann-Whitney U test. For multiple differences between groups, the Kruskal-Wallis test was used followed by Dunn's test with Bonferroni adjustment at p≤0.05 (Dunn-Bonferroni). Any value of p≤0.05 was considered significant. All tests were performed with the IBM SPSS statistics version 23 program.”

Comment. The finding that glucose was significantly elevated in patients with DM is what defines DM in the first place. Why is this reported? To confirm the diagnosis?

Answer. Thank you for the pertinence of the comment about including the increased glucose levels. The determination was to define the presence of diabetes mellitus. We consider that, although the glucose alteration is apparent, the glucose result in both groups should be visible in the document

Comment. Why wasn’t serum insulin level also reported? Did all the patients have Type I and/or Type II diabetes? Does this matter?

Answer. Patients included with diabetes have type 2 diabetes

Comment. The use of the lowercase letters in the tables as indicators of significance is unclear. Specifically, what is different from what? For the creatinine level, there is an ‘a’ and a ‘b’ for the Stage 2 value in the No DM group, so does this mean it is different from both the Stage 1 and Stage 3 patients?

Answer.   The use of lowercase letters is used as an indicator of the significant differences found in the post-hoc analysis between the different stages of CKD. Letter (a), (b) and (c) indicates significant difference compared to stage 1, 2 and 3 respectively.

Comment. Why does the numbering scheme for citations switch to roman numerals?

Answer.  I offer a vast apology; due to an oversight, I did not realize that from the discussion, the bibliographical references changed from progressive numbers to Roman numerals. I will be much more careful that in the discussion, the numbers are in the proper sequence

Reviewer 2 Report

The abstract is quite chaotic. Not all information is needed. Some are obvious and well known, such as the fact that PTH levels are higher and eGFR lower in CKD. The authors wrote that eGFR decreased as kidney disease progressed. After all, it is evident that the reduction of eGFR is one of the critical criteria for classifying CKD. The summary should include information about the results related to the mainstream of work, i.e., oxidative stress. The summary in the abstract is also apparent; in fact, everyone knows that early diagnosis and treatment of CKD slow the progression and achievement of ESRD.

The entire summary should be rewritten. Moreover, the abbreviations NS and ESRD appear for the first time in the abstract and are not explained here. It needs to be corrected.

Introduction.
The authors described the diagnosis of CKD well. However, strange wording already appeared in the second paragraph. For example, "Patients in stages 1 and 2 are at high risk of GFR reduction." After all, the classification of these groups is usually based on reduced eGFR. This sentence should be deleted.
The next sentence also arouses my objection. The authors wrote that in stage 3, anemia might occur due to the destruction of red blood cells. What kind of erythrocyte destruction do the authors mean when mentioning CKD stage 3? Anemia in these stages is primarily due to erythropoietin deficiency and other factors. Destruction as a result of toxemia occurs in more advanced stages. I recommend the article: Podkowińska, A .; Formanowicz, D. Chronic Kidney Disease as a Cardiovascular Disease Related to Oxidative Stress and Inflammation Antioxidants 2020, 9, 752. https://doi.org/10.3390/antiox9080752. It has been well explained here.

The next sentence is wrong: "Currently, the diagnosis of CKD is made by blood urea and serum creatinine (sCr) levels; however, it has been shown that sCr lacks sufficient predictive value [7] ". After all, it is well known today we do not rely on creatinine concentration due to its dependence, for example, on body weight, age, and gender; hence we use eGFR. The formulated sentence, in this way, misleads the reader.

Next sentence: It is possible that the inflammatory and oxidative states actively participate in kidney damage in the early stages. It is also a well-known issue; hence the word "possible" should be removed.

The description of oxidative stress, which is the article's main topic, is very economical, but I think it should be expanded. I recommend the available articles again, incl. the above-mentioned: "Podkowińska, A .; Formanowicz, D. Chronic kidney disease as a cardiovascular disease mediated by oxidative stress and inflammation. Antioxidants 2020, 9, 752. https://doi.org/10.3390/antiox9080752. "

The description of the research group is correct. However, I recommend moving Table 1 here and removing it from the results. These are not the results but the characteristics of the group. Besides, going back to table 1 - it is unclear and difficult to understand; I recommend changing it.
In my opinion, the methodology for determining individual parameters is described correctly.

Results

The first sentence is strange. “Eighty-eight patients were enrolled; eighty-eight were without diabetes and one hundred
with DM. Eighty-one men and one hundred and seven women. The average age was 50.1 ± 22.02 years ."

Since 88 patients were included, something is wrong here because the authors wrote that there were 88 patients without diabetes and 100 with diabetes. And then, they reported that 81 were men and 107 were women. Such a presentation, apart from mistakes, gives little to the reader. It is better to indicate what was the mean age in each group and what was the gender distribution in the group with and without diabetes.
Were there differences between the groups in this regard?

The presentation of the results of basic research also raises reservations. What for the authors wrote that patients with DM had higher glucose levels and sCR increased as the disease progressed? Next, the authors write about the difference between the diabetic and non-diabetic groups at p = 004. However, this difference is not only due to the presence of diabetes but to how many patients were recruited to each group at what stage of CKD (1 or 3). As I wrote at the beginning, the mentioned results descriptions do not add much. For example, see the sentence: "Parathyroid hormone levels increased as kidney disease progressed in patients without DM (p=0.03) and was higher in patients with DM (NS)." What gives this sentence to the readers?
Table 2 is very good. Authors should focus on the matters of others rather than scoring the obvious in their results: Eg, BMI, cholesterol, and fractions. This table can be moved to the group characteristics. The assessments made characterize the study group but are not the results. The parameters of oxidative stress are the results of this study. The authors should characterize the parameters of oxidative stress in the context of healthy people. Otherwise, it is not known how to interpret the sentence: “The pro-inflammatory cytokines IL-6 and TNF-α were identical in the three early stages of the disease in patients with and without diabetes. groups (p <0.01) ".
Was the level of cytokines higher or lower than in healthy people? Was there a significant correlation between inflammation and the assessed oxidative status parameters?

Moreover, the authors are imprecise in many places. e.g., "However, the levels of NO in DM patients were similar in all three phases." What phases? There should be stages instead of stages.

Discussion.
Authors in many places "deviate" from care. Why describe the calcium-phosphate metabolism when it is not the main topic? Or they are focusing on eGFR.

Besides, I do not understand the abbreviations here, like [XXXVI), [XXXViii], and [XXXiV]. What is it about?

In general, the discussion is too long; I recommend shortening it, focusing on oxidative stress, and explaining the observed differences between groups and stages. By explanation, I mean looking for the cause, not just a description. Besides, I recommend the reformulation of the summary and conclusions.

Author Response

Reviewer 2

Comment. The abstract is quite chaotic. Not all information is needed. Some are obvious and well known, such as the fact that PTH levels are higher and eGFR lower in CKD. The authors wrote that eGFR decreased as kidney disease progressed. After all, it is evident that the reduction of eGFR is one of the critical criteria for classifying CKD. The summary should include information about the results related to the mainstream of work, i.e., oxidative stress. The summary in the abstract is also apparent; in fact, everyone knows that early diagnosis and treatment of CKD slow the progression and achievement of ESRD.

Answer. Thank you for your comment. You are very suitable; the summary is chaotic and includes apparent results. The abstract has been redrafted; we hope the current version will be more relevant.

Abstract

Chronic kidney disease (CKD) is a risk factor for end-stage renal disease (ESRD) progression. Early diagnosis of kidney disease in the early stages can reduce complications. Alterations in renal function represent a complication of diabetes mellitus (DM). The mechanisms underlying the progression of CKD in diabetes could be associated with oxidative and inflammatory processes. The study aimed to evaluate the state of inflammation and oxidative stress (OS) on the progression of CKD in the early stages in patients with and without type 2 diabetes mellitus (T2DM). An analytical cross-sectional study was carried out in patients with CKD in early stages (1, 2, 3) with and without T2DM. The ELISA method determined the expression of the pro-inflammatory cytokines IL-6 and TNF-α. Lipoperoxides (LPO), nitric oxide (NO), and superoxide dismutase activity. (SOD), colorimetric methods determined glutathione peroxidase (GPx) and total antioxidant capacity (TAC). Patients with CKD and T2DM had significantly decreased antioxidant defenses for SOD (p<0.01), GPx (p<0.01), and TAC (p<0.01) compared to patients without T2DM. Consequently, patients with T2DM had higher concentrations of oxidant markers, NO (p<0.01), inflammation markers, IL-6 (p<0.01), and TNF-α than patients without T2DM. CKD stages were not related to oxidative, antioxidant, and inflammatory marker outcomes in T2DM patients. Patients without T2DM presented an increase in SOD (p=0.04) and a decrease in NO (p<0.01) when the stage level of CKD increased. In conclusion, patients with T2DM present higher levels of oxidative and inflammatory markers accompanied by a decrease in antioxidant defense. However, these oxidative status markers were associated with CKD stage progression in patients without T2DM. NO and SOD markers could help detect the early stages of CKD in patients who have not yet developed metabolic comorbidities such as T2DM

Comment. The entire summary should be rewritten. Moreover, the abbreviations NS and ESRD appear for the first time in the abstract and are not explained here. It needs to be corrected.

Answer. The full summary was rewritten. The abbreviation NS was removed, and end-stage renal disease is ESRD

Introduction.

Comment. The authors described the diagnosis of CKD well. However, strange wording already appeared in the second paragraph. For example, "Patients in stages 1 and 2 are at high risk of GFR reduction." After all, the classification of these groups is usually based on reduced eGFR. This sentence should be deleted.

Answer. “Patients in stages 1 and 2 are at high risk of GFR reduction” was delated

Comment. The next sentence also arouses my objection. The authors wrote that in stage 3, anemia might occur due to the destruction of red blood cells. What kind of erythrocyte destruction do the authors mean when mentioning CKD stage 3? Anemia in these stages is primarily due to erythropoietin deficiency and other factors. Destruction as a result of toxemia occurs in more advanced stages. I recommend the article: Podkowińska, A .; Formanowicz, D. Chronic Kidney Disease as a Cardiovascular Disease Related to Oxidative Stress and Inflammation Antioxidants 2020, 9, 752. https://doi.org/10.3390/antiox9080752. It has been well explained here.

Answer. You are absolutely right. The writing on anemia was very unlucky. The wording about kidney disease involvement in erythrocytes was changed to the discussion. We fully accept your suggestions. The text was as follows

In the present study, Hb decreased as the stage of kidney disease progressed. Various factors could explain this; we must consider that erythrocytes play a fundamental role in maintaining the oxidative status of the blood. There are several mechanisms by which oxidative damage and inflammation cause a decrease in Hb by iron deficiency in patients with CKD, including the reduction in antioxidants and oxidation of erythrocyte membrane phospholipids. Other mechanisms are the absolute deficiency of iron characterized by reduced or absent stores. Functional iron deficiency is another mechanism where iron availability to be incorporated into erythroid precursors is insufficient due to increased hepcidin levels.

Furthermore, CKD predisposes patients to vitamin and mineral deficiencies, which contribute to anemia, CVD, and metabolic imbalances. There are longitudinal studies that show that as CKD progresses, Hb levels decrease until anemia occurs. Oxidative damage causes premature red blood cell aging by increased production of hepcidin, reduced production of the iron transport protein transferrin and its receptor, and induction of resistance to erythropoietin. Therefore, the combination of factors may be responsible for the decrease in Hb in patients with early stages of CKD included in the study.

Comment. The next sentence is wrong: "Currently, the diagnosis of CKD is made by blood urea and serum creatinine (sCr) levels; however, it has been shown that sCr lacks sufficient predictive value [7] ". After all, it is well known today we do not rely on creatinine concentration due to its dependence, for example, on body weight, age, and gender; hence we use eGFR. The formulated sentence, in this way, misleads the reader.

Answer. The sentence was formulated as follows: “Currently, the diagnosis of CKD is made by blood urea and serum creatinine (sCr) levels; however, sCr concentration is not very reliable due to its dependence on the patient's body weight, age, and gender. However, GFR is considered the gold standard for the evaluation of kidney function

Comment. Next sentence: It is possible that the inflammatory and oxidative states actively participate in kidney damage in the early stages. It is also a well-known issue; hence the word "possible" should be removed.

Answer. The word “possible” was removed

Comment. The description of oxidative stress, which is the article's main topic, is very economical, but I think it should be expanded. I recommend the available articles again, incl. the above-mentioned: "Podkowińska, A .; Formanowicz, D. Chronic kidney disease as a cardiovascular disease mediated by oxidative stress and inflammation. Antioxidants 2020, 9, 752. https://doi.org/10.3390/antiox9080752."

Answer. Thank you for your comment, you are undoubtedly right. The writing of the mechanisms of action of oxidative stress was strengthened with the support of the recommended scientific article

“Oxidative stress (OS) is defined as the imbalance between reactive oxygen species (ROS), reactive nitrogen species (RNS), and free radicals as oxidants. Oxidants are unstable molecules that promote oxidation reactions with other molecules, such as proteins, lipids, and DNA, to stabilize themselves. ROS includes free radicals with an unpaired electron on an oxygen atom, eg, hydroperoxides (HO2•), superoxide anion radical (O2•−), and hydroxyl radical (HO•). Redox signaling agents have oxidizing properties such as hydrogen peroxide (H2O2) or singlet oxygen (1O3). OS reactions also involve oxygen compounds such as non-radical compounds (peroxynitrite (ONOO-) and hypochlorous acid (HOCl). The metabolic imbalance of OS leads to depletion of endogenous and exogenous antioxidants, abnormal protein function, damage to deoxyribonucleic acid (DNA), enzymes, and finally, irreversible cell damage. In diabetic-CKD, the production of ROS in the kidney is mainly due to increased expression and activity of NADPH oxidase (NOX). Increased NOX-mediated ROS in the kidney are considered mediators of glomerulosclerosis, endothelial dysfunction, development of albuminuria, and disruption of glomerular hemodynamics.”

Comment. The description of the research group is correct. However, I recommend moving Table 1 here and removing it from the results. These are not the results but the characteristics of the group. Besides, going back to table 1 - it is unclear and difficult to understand; I recommend changing it.
In my opinion, the methodology for determining individual parameters is described correctly.

Answer. Table 1 was modified and installed in the material and methods section; we hope it will be more relevant in that location

Table 1. Demographic data

No DM

T2DM

p

                   Age

50±22

62±14

<0.01*

                   Gender

Male (n)

39

42

0.431

Female (n)

49

58

Stage

1 (n)

8

10

0.962

2 (n)

34

37

3 (n)

46

53

Hypertension

Hypertensive (n)

71

65

<0.01**

No hypertensive (n)

17

35

Alcohol

Alcohol consumption (n)

13

7

0.068

No alcohol consumption (n)

75

93

Tobacco consumption

Smoker (n)

38

55

0.917

No smoker (n)

50

45

The results are expressed in numbers and mean ± standard deviation. DM: diabetes mellitus. *Significant for the t-Student test. **Significant for Chi2 test

Results

Comment. The first sentence is strange. “Eighty-eight patients were enrolled; eighty-eight were without diabetes and one hundred
with DM. Eighty-one men and one hundred and seven women. The average age was 50.1 ± 22.02 years."

Answer. I apologize for the error. Text was changed

“One hundred patients with T2DM and eighty-eight without DM were included. Patients with T2DM were older and had less hypertension than patients without DM. Table 1”

Comment. Since 88 patients were included, something is wrong here because the authors wrote that there were 88 patients without diabetes and 100 with diabetes. And then, they reported that 81 were men and 107 were women. Such a presentation, apart from mistakes, gives little to the reader. It is better to indicate what was the mean age in each group and what was the gender distribution in the group with and without diabetes. Were there differences between the groups in this regard?

Answer. The changes can be seen in Table 1. We hope that the table has become clearer

Comment. The presentation of the results of basic research also raises reservations. What for the authors wrote that patients with DM had higher glucose levels and sCR increased as the disease progressed? Next, the authors write about the difference between the diabetic and non-diabetic groups at p = 004. However, this difference is not only due to the presence of diabetes but to how many patients were recruited to each group at what stage of CKD (1 or 3). As I wrote at the beginning, the mentioned results descriptions do not add much. For example, see the sentence: "Parathyroid hormone levels increased as kidney disease progressed in patients without DM (p=0.03) and was higher in patients with DM (NS)." What gives this sentence to the readers?
Table 2 is very good. Authors should focus on the matters of others rather than scoring the obvious in their results: Eg, BMI, cholesterol, and fractions. This table can be moved to the group characteristics. The assessments made characterize the study group but are not the results. The parameters of oxidative stress are the results of this study. The authors should characterize the parameters of oxidative stress in the context of healthy people. Otherwise, it is not known how to interpret the sentence: “The pro-inflammatory cytokines IL-6 and TNF-α were identical in the three early stages of the disease in patients with and without diabetes. groups (p <0.01) ".
Was the level of cytokines higher or lower than in healthy people? Was there a significant correlation between inflammation and the assessed oxidative status parameters?

Answer. The results section was modified, and the last column of Table 3 was eliminated because it confused what it was referring to. The results were broken down in Table 4, which gives more relevance to the results. We apologize for not taking advantage of the results in the initial document

Table 4. Inflammatory and oxidative stress markers between patients with and without DM

No DM

DM

p

Oxidants

Lipoperoxides (mM)

29.3±22.40

26.58±24.25

0.1

Nitric oxide (µg/mL)

1921.99±1328.03

2470.63±422.83

<0.01*

Antioxidants

Superoxide dismutase (U/L)

17.24±7.02

14.64±6.10

<0.01*

Glutathione peroxidase (nmol/min/mL)

10.79±0.38

10.86±0.30

<0.01*

Total Antioxidant Capacity  (µM)

1.57±0.63

1.24±0.44

<0.01*

Pro-inflammatory cytokines

Interleukine 6 (pg/mL)

45.81±11.25

57.23±23.37

<0.01*

Tumor Necrosis Factor α  (pg/mL)

6.31±2.01

6.932.09

<0.01*

The results are expressed as mean ± standard deviation. *M-W: Mann-Whitney U test.

Comment. Moreover, the authors are imprecise in many places. e.g., "However, the levels of NO in DM patients were similar in all three phases." What phases? There should be stages instead of stages.

Answer.  Changed word phases to stages

Discussion.
Comment. Authors in many places "deviate" from care. Why describe the calcium-phosphate metabolism when it is not the main topic? Or they are focusing on eGFR.

Answer. You are right, calcium-phosphorus metabolism was removed from the discussion. More relevance was given to changes in the inflammatory and oxidative state with kidney function

Comment. Besides, I do not understand the abbreviations here, like [XXXVI), [XXXViii], and [XXXiV]. What is it about?

Answer. I offer a vast apology; due to an oversight, I did not realize that from the discussion, the bibliographical references changed from progressive numbers to Roman numerals. I will be much more careful that in the discussion, the numbers are in the proper sequence

Comment. In general, the discussion is too long; I recommend shortening it, focusing on oxidative stress, and explaining the observed differences between groups and stages. By explanation, I mean looking for the cause, not just a description.

Answer. The discussion was shortening. We hope it is more focused and better written

Besides, I recommend the reformulation of the summary and conclusions.

Answer. Reformulated summary and conclusions

Abstract

Chronic kidney disease (CKD) is a risk factor for end-stage renal disease (ESRD) progression. Early diagnosis of kidney disease in the early stages can reduce complications. Alterations in renal function represent a complication of diabetes mellitus (DM). The mechanisms underlying the progression of CKD in diabetes could be associated with oxidative and inflammatory processes. The study aimed to evaluate the state of inflammation and oxidative stress (OS) on the progression of CKD in the early stages in patients with and without type 2 diabetes mellitus (T2DM). An analytical cross-sectional study was carried out in patients with CKD in early stages (1, 2, 3) with and without T2DM. The ELISA method determined the expression of the pro-inflammatory cytokines IL-6 and TNF-α. Lipoperoxides (LPO), nitric oxide (NO), and superoxide dismutase activity. (SOD), colorimetric methods determined glutathione peroxidase (GPx) and total antioxidant capacity (TAC). Patients with CKD and T2DM had significantly decreased antioxidant defenses for SOD (p<0.01), GPx (p<0.01), and TAC (p<0.01) compared to patients without T2DM. Consequently, patients with T2DM had higher concentrations of oxidant markers, NO (p<0.01), inflammation markers, IL-6 (p<0.01), and TNF-α than patients without T2DM. CKD stages were not related to oxidative, antioxidant, and inflammatory marker outcomes in T2DM patients. Patients without T2DM presented an increase in SOD (p=0.04) and a decrease in NO (p<0.01) when the stage level of CKD increased. In conclusion, patients with T2DM present higher levels of oxidative and inflammatory markers accompanied by a decrease in antioxidant defense. However, these oxidative status markers were associated with CKD stage progression in patients without T2DM. NO, and SOD markers could be helpful in detecting the early stages of CKD in patients who have not yet developed metabolic comorbidities such as T2DM

In conclusion, oxidative imbalance was evidenced between patients with and without T2DM. The presence of OS in T2DM was characterized by an increase in oxidative markers and a decrease in antioxidant defenses (SOD, TAC, GPx). Only patients without T2DM showed a significant increase in the activity of the antioxidant enzyme SOD and a substantial decrease in NO when the CKD stage increased. In the present study, SOD and NO markers changed during the early stages of CDK similarly to those of clinical parameters Hb, hematocrit, and sCr, frequently used to assess CDK progression. NO, and SOD markers could help detect the early stages of CKD in patients without T2DM. Detecting and treating early kidney disease offers an invaluable opportunity to activate the different actions available to delay the progression of the disease and prevent its evolution to ESRD.

Round 2

Reviewer 1 Report

I reviewed the initial submission. Thank you for taking the time to address each of my comments thoughtfully and thoroughly.  I have no further comments/suggestions.

Reviewer 2 Report

Dear Authors,

thank you for considering all my comments. It seems to me that our joint work :) was successful.
This article is well prepared and, in my opinion, worthy of approval and publication.

Only one last suggestion. I believe that before the publication itself, the Authors should correct some labeling in Tables 2 and 3; I mean the number of patients; not n-8 but n=8, and this is valid for all CKD stages.

Best  regards
